# Co-Zn-MOFs Derived N-Doped Carbon Nanotubes with Crystalline Co Nanoparticles Embedded as Effective Oxygen Electrocatalysts

**DOI:** 10.3390/nano11020261

**Published:** 2021-01-20

**Authors:** Wendi Zhang, Xiaoming Liu, Man Gao, Hong Shang, Xuanhe Liu

**Affiliations:** School of Science, China University of Geosciences (Beijing), Beijing 100083, China; zhangwd@cugb.edu.cn (W.Z.); liuxm@cugb.edu.cn (X.L.); goyeman@outlook.com (M.G.); shanghong@cugb.edu.cn (H.S.)

**Keywords:** metal-organic frameworks, N-doped carbon, oxygen reduction reaction

## Abstract

The oxygen reduction reaction (ORR) is a crucial step in fuel cells and metal-air batteries. It is necessary to expand the range of efficient non-precious ORR electrocatalysts on account of the low abundance and high cost of Pt/C catalysts. Herein, we synthesized crystalline cobalt-embedded N-doped carbon nanotubes (Co@CNTs-T) via facile carbonization of Co/Zn metal-organic frameworks (MOFs) with dicyandiamide at different temperatures (t = 600, 700, 800, 900 °C). Co@CNTs- 800 possessed excellent ORR activities in alkaline electrolytes with a half wave potential of 0.846 V vs. RHE (Reversible Hydrogen Electrode), which was comparable to Pt/C. This three-dimensional network, formed by Co@CNTs-T, facilitated electron migration and ion diffusion during the ORR process. The carbon shell surrounding the Co nanoparticles resulted in Co@CNTs-800 being stable as an electrocatalyst. This work provides a new strategy to design efficient and low-cost oxygen catalysts.

## 1. Introduction

Excessive consumption of fossil fuels gives rise to an energy dilemma as well as environmental concerns. Therefore, it is necessary to expand new energy storage technologies, such as electrochemical energy storage [1,2]. Metal-air batteries and fuel cells are promising for energy conversion and storage devices [3,4]. Nevertheless, the sluggish kinetics of cathodic oxygen reduction reactions (ORR) needs to be urgently addressed [5]. Pt-based catalysts are effective as catalysts in ORR [6,7]. However, their high cost and scarcity make the development of other types of ORR electrocatalysts with low price and high activity extremely important, such as metal-free [8,9] and transition metal electrocatalysts [10,11].

Metal-free electrocatalysts like carbon-based catalysts have been widely studied for their good stability. However, the average electrocatalytic performance has limited their applications [12,13]. The 3d transition metal electrocatalysts (such as Ni, Fe, and Co) are considered to have promising catalytic activity due to their unique electronic structure [14,15]. Co-based catalysts are attractive due to the higher stability on account of their immunity to the Fenton reaction [16,17]. Heteroatom doping (such as N, S, P) [18,19], introduction of defects [20,21], and morphological control [22] were considered as effective methods to improve the electrocatalytic performance. Metal-organic frameworks (MOFs) are of considerable interest in terms of alternative active metal centers, regulable structures, and high specific surface area [23,24]. MOFs are considered to be eminent precursors in obtaining porous nanocomposites having a variety of morphology features and chemical components by adjusting the precursors and calcination conditions [25,26].

Zn/Co-based MOFs are promising precursors to fabricate heteroatom-doped metal-supported carbon materials but more defects were expected to be created by Zn volatilization at high temperature. However, Zn volatilizing at high temperature could prevent the accumulation of other metal ions [27,28,29]. The morphology and structure of the catalysts also play a significant role in enhancing their electrocatalytic activity. Herein, we fabricated a three-dimensional network composed of N-doped carbon nanotubes with crystalline Co nanoparticles, embedded, by calcination of Co/Zn MOF and dicyandiamide (DCD) as the precursor. The three-dimensional network, termed Co@CNTs, was constructed by regulating the annealing temperature and the mass ratios of Co/Zn MOF and DCD. The obtained Co@CNTs composite showed a nanotube structure, while crystalline Co nanoparticles were embedded at the end of the CNTs, surrounded by the carbon shell. The Co@CNTs prepared at 800 °C (Co/Zn MOF:DCD = 1:1) exhibited more densely packed and uniform nanotubes and better ORR performance. The half-wave potential was 0.846 V vs. RHE and showed good electrochemical stability, comparable to commercial Pt/C. This work could provide a synthetic method for economical and durable electrocatalysts with three-dimensional networks for ORR.

## 2. Materials and Methods

### 2.1. Preparation of Co-Zn-BDC

All chemical reagents were bought from Aladdin Reagent (Aladdin, Shanghai, China) and used directly in the present experiments. The Co-Zn-BDC was constructed by ultrasound [30]. First, 0.75 mmol 1,4-benzenedicarboxylic acid (H_2_BDC) was dissolved to 36 mL mixed solution of DMF, ethanol, and deionized water with a volume ratio of 16:1:1. Subsequently, 0.75 mmol CoCl_2_∙6H_2_O and 0.75 mmol ZnCl_2_ were added, respectively. After the Co^2+^ and Zn^2+^ salts were fully dissolved, 0.8 mL of triethylamine (TEA) was then added to the solution. After stirring for 5 min the above mixture underwent 8 h ultrasonication (40 kHz). The resulting pink precipitate was collected by centrifugation and rinsed with ethanol and water, respectively, followed by vacuum freeze-drying. Co-BDC was prepared by the same method without the introduction of Zn ions.

### 2.2. Preparation of Co@CNTs-T and Control Samples

The mixture of Co-Zn-BDC and DCD with mass ration of 1:1 underwent calcination at high temperatures (600, 700, 800 and 900 °C) for 2 h under nitrogen protection at a heating rate of 5 °C min^−1^. The products were labelled as Co@CNTs-600, Co@CNTs-700, Co@CNTs-800, and Co@CNTs-900, respectively. The schematic synthetic process is depicted in Scheme 1. Co-MOF-800 was prepared by annealing Co-BDC at the temperature of 800 °C, and Co-Zn-MOF-800 was synthesized by annealing CO-Zn-BDC at the temperature of 800 °C.

### 2.3. Sample Characterization Methods

Scanning electron microscopy (SEM, ZEISS MERLIN Compact, 20 Kv, Jena, Germany) and field emission transmission electron microscope (TEM, FEI G2 F30 200 kV, Hillsboro, OR, USA) were carried out to observe the micro-morphology of Co@CNTs-T. Power X-ray diffraction (XRD, D8 SmartLab (Karlsruhe, Germany) with Cu Kα radiation (λ = 0.15418 nm)) patterns were characterized to obtain crystal structures. X-ray photoelectron spectroscopy (XPS, Escalab 250Xi, Waltham, America) was used to probe the chemical states of surface elements. Fourier transform infrared spectroscopy (FTIR IRTracer-100, Tokyo, Japan) was employed to analyze chemical and structural information. Thermogravimetric analysis (TGA, NETZSCH-200F3, Selbu, Germany) was used to analyse thermal stability. ICP-MS Agilent 7700 (Palo Alto, CA, USA) analyzed the composition of metal elements. Raman (Renishaw, London, UK) was used to analyse the purity of materials. BET (ASAP 2020, Norcross, GA, USA) analysed the specific surface area and pore size. The Organic Element Analyzer (EA Vario EL cube, Bonn, Germany) was used to analyse the CHN of materials.

### 2.4. Electrochemical Measurement

Typically, a homogeneous black solution was formed by fully dispersing a 3 mg sample of Co@CNTs-T, 7 mg carbon black, and 1.25 mL ethanol under ultrasound for 3 h. Then 36 μL of the ink and 6 μL Nafion solution (0.5 wt%) were coated on the pre-polished rotating ring-disk electrode (RRDE, 4 mm in diameter) followed by drying at room temperature. A homogeneous Pt/C ink solution was formed by fully dispersing 3 mg commercial Pt/C and 1.25 mL ethanol under ultrasound for 3 h. Then 36 μL of the ink and 6 μL Nafion solution (0.5 wt%) were coated on the pre-polished RRDE electrode followed by drying at room temperature. The RRDE electrode covered with catalyst was used as the working electrode and the saturated Ag/AgCl electrode was used as the reference electrode. The graphite electrode was used as the counter electrode and the electrolyte was 0.1 M KOH solution (N_2_ or O_2_ were pumped into 0.1 M KOH solution for half an hour respectively). Cyclic voltammetry was used at the scan rate 50 mV s^−1^ and linear sweep voltammetry at the scan rate of 5 mV s^−1^ with a rotating speed of 1600 rpm. Besides chronoamperometry at 0.7 V (vs. RHE) testing of the prepared sample was conducted in the RRDE test system.

## 3. Results

### 3.1. Structural Properties of the Co@CNTs-T

Scanning electron microscope (SEM) was used to investigate the morphology of Co- Zn-BDC, and the SEM image shown in Appendix A displays nanosheet morphology of Co-Zn-BDC. The XRD pattern of Co-Zn-BDC is shown in Appendix A, the peaks appearing at 8.8°, 15.8°, 17.8°, 28.8°, 30.8° demonstrated its crystal structure, which was consistent with previous reports [31,32]. Fourier transform infrared spectroscopy (FTIR) was carried out to determine the chemical and structural information. The FTIR data of Co-Zn-BDC is shown in Appendix A. The strong absorption peaks of H_2_BDC (black) appearing at 1673 and 1424 cm^−1^ originate from the asymmetric and symmetric stretching vibrations of crude carboxylate groups (–COOH). For Co-Zn-BDC (red), the peaks were red shifted to 1573 and 1362 cm^−1^, attributed to the coordination of carboxylate organic ligands and metal centers [33]. Then, inductively coupled plasma-mass spectrometry (ICP-MS) was used to determine the Co/Zn content of Co-Zn-BDC; the result indicated that Co-Zn-BDC contained 76.73 wt% Co and 23.27 wt% Zn, respectively. Besides, thermogravimetric analysis was performed to analyze the thermal stability of Co-Zn-BDC. As shown in Appendix A, Co-Zn-BDC remained stable with a slight decrease in weight before 400 °C. As the temperature rose, the weight was lost obviously due to pyrolysis.

The morphology of all Co@CNTs-T was also studied by SEM. The SEM images are shown in Figure 1a–d. Particles with a few nanotubes are seen from the SEM image of Co@CNTs-600 in Figure 1a. With the temperature rising in the range of 600–800 °C, more nanotubes arose and were distributed evenly. They were connected to constitute a three-dimensional network structure, which was beneficial to electron transfer in the electrocatalytic process [34]. Nevertheless, when the annealing temperature reached 900 °C, there were no visible nanotubes and only large-sized nanoparticles with diameter of 50–200 nm could be seen. We concluded that higher annealing temperatures were not suitable to form nanotubes. Besides, the mass ratio of Co-Zn-BDC/DCD was of crucial importance in producing uniform nanotubes. The SEM images for Co@CNTs-800 by annealing Co-Zn-BDC/DCD with different mass ratios (2:1 and 1:2) are shown in Appendix A. It can be seen that no visible nanotubes were produced. The mass ratio of 1:1 was most suitable for the formation of a three-dimensional network structure.

Transmission electron microscopy (TEM) was used to further study the morphology and phase composition of Co@CNTs-800. The TEM image in Figure 2a revealed that the prepared nanotubes were of hundreds of nanometers in length with diameter of ~50 nm, the nanoparticles were formed at the end of the nanotubes and the diameters of the nanoparticles were about 20–60 nm. The high-resolution TEM image (Figure 2b) displayed nanoparticles with spacing of 0.20 nm, which were indexed as the Miller indices (112) of Co. The results revealed that the crystalline Co nanoparticles were embedded in nanotubes, the nanoparticles were surrounded with a carbon layer, and the lattice fringe spacing of the carbon layer was 0.31 nm, assigned to the (002) plane of graphitic carbon [35]. The elemental mapping images in Figure 2c–f indicated that C, Co, and N elements were distributed evenly in the nanotubes. It was noted that more Co elements were concentrated in the nanoparticles region.

X-ray Diffraction (XRD) was performed to show the chemical phase composition and XRD patterns of Co@CNTs-600, Co@CNTs-700, Co@CNTs-800, and Co@CNTs-900, displayed in Figure 3a. The broad peak located at ~26.1° corresponds to the C (002) plane. The intense peaks of Co@CNTs-600 appearing at 31.8°, 34.4° and 36.3° were successively assigned to the (100), (002), and (101) planes of ZnO, consistent with the previously reported hexagonal ZnO (JCPDS no 36-1451). The diffraction peaks appearing at 44.2°, 51.5°, and 75.8° were assigned to the (111), (200) and (220) (planes of cubic Co (JCPDS no 15-0806) [36]. These intense and sharp peaks are indicative of high crystallinity. No other diffraction peaks were seen in the XRD patterns, which demonstrated that no other phases had appeared. The characteristic diffraction peaks of ZnO disappeared in Co@CNTs-700, Co@CNTs-800, and Co@CNTs-900. The ICP-MS analysis revealed that Co@CNTs-800 contained 97.88 wt% Co and 2.12 wt% Zn, respectively. The Zn species in Co-Zn-BDC volatilized with the higher annealing temperature on account of the low boiling point of Zn during the carbonization [37].

Raman spectra of Co@CNTs-T and Co@C-800 were collected to evaluate the catalyst. As shown in Figure 3b and Appendix A, the peaks appearing at 1350 and 1584 cm^−1^ were attributed to disordered (D band) and graphite (G band) carbon. The I(D)/I(G) ratio of Co@CNTs-800 was 1.02, which was higher than Co@CNTs-600(1.00), Co@CNTs-700(0.99), Co@CNTs-900(0.74), and Co@C-800(0.91). More defects formed in Co@CNTs-800 which were expected to facilitate the electrocatalytic performance [38]. The Organic Element Analyzer (EA) test was used to analyze the content of C, H, and N. As shown in Appendix A, the results showed that the N contents of Co@CNTs-600, Co@CNTs-700, Co@CNTs-800, and Co@CNTs-900 were 2.75%, 3.01%, 1.87%, and 1.25%. Combined with the Raman results, we concluded that both N content and defects play an important role in enhancing ORR performance [39].

X-ray photoelectron spectroscopy (XPS) measurements were then used to investigate chemical configurations and the valence state of Co@CNTs-800. XPS survey spectra in Figure 3c indicated C 1s, N 1s, O 1s, and Co 2p peaks with no other signals. The contents of C, O, N, and Co were 87.7, 5.33, 4.40, and 2.47 wt%, respectively. The C 1s XPS spectrum in Figure 3d confirmed four kinds of C species: C=C (284.4 eV), C-C (285.2 eV), C=N (286.1 eV), and C-N (288.5 eV) [40,41]. The N 1s spectra in Figure 3e can be fitted deconvoluted as four bands at 398.5, 399.7, 400.7, and 401.7 eV, which could be attributable to four types of N species: pyridine-N, pyrrolic-N, graphitic-N, and pyridine-N-O [42]. Their proportions were 1.84, 0.58, 1.06, and 0.93%, respectively (Appendix A). Previous studies indicated that pyridine-N and graphitic-N contributed to boosting electrocatalytic activity in ORR [43,44,45]. We concluded that Co@CNTs-800 had potential as an effective ORR catalyst due to the high proportion of pyridine-N and graphitic-N. The high-resolution Co 2p spectrum is shown in Figure 3f. The binding energies of 778.5 and 793.4 eV were due to 2p_3/2_ and 2p_1/2_ of Co^3+^. The binding energies of 780.3 and 796.1 eV were assigned to 2p_3/2_ and 2p_1/2_ of Co^2+^, and peaks located at 784.8 and 802.5 eV were characteristic satellite peaks, consistent with previous reports [46,47]. The different valence states of the cobalt active species on the metal nanoparticles were also beneficial in enhancing the electrocatalytic activity [48].

The N_2_ adsorption/desorption test was used to explore the surface area and pore structures of Co@CNTs-800. As shown in Figure 4a, the Brunner–Emmet–Teller specific surface area was revealed to be 52.7 m^2^ g^−1^. The N_2_ adsorption/desorption curve exhibited a characteristic type IV isotherm and a distinct hysteresis loop at relative high pressure, demonstrating their mesoporous structure. The pore size distribution curve in Figure 4b was used to confirm the mesoporous feature. The pore diameters located at 4.0, 7.6, 10.7, and 28.9 nm indicated the presence of mesopore, beneficial to the rapid mass transfer process.

### 3.2. Electrocatalytic Performance

The electrocatalytic activities of Co@CNTs-600, Co@CNTs-700, Co@CNTs-800 and Co@CNTs-900 were studied by cyclic voltammetry (CV) in 0.1 M KOH with saturated O_2_ or saturated N_2_ at 50 mV·s^−1^. Appendix A displays CV curves of all the electrode materials. No distinct redox peaks in N_2_ saturated electrolyte can be observed for all the electrode materials, while obvious cathodic peaks appear when in O_2_-saturated electrolyte. The cathodic peak for Co@CNTs-800 (0.785 V vs. RHE, all potentials refer to RHE herein), was more positive than those of Co@CNTs-600, Co@CNTs-700, and Co@CNTs-900 (0.677, 0.734, and 0.725 V respectively). The CV results demonstrated that Co@CNTs-800 showed better catalytic activity.

The electrocatalytic ORR behavior of Co@CNTs-600, Co@CNTs-700, Co@CNTs-800, and Co@CNTs-900 were further investigated by linear sweep voltammetry (LSV) measurements. Figure 5a shows the LSV curves of all the electrode materials. The half wave potentials were 0.772, 0.817, and 0.796 V for Co@CNTs-600, Co@CNTs-700, and Co@CNTs-900, respectively. For Co@CNTs-800, the half wave potential was the most positive (0.846V), which was equivalent to that of the commercial Pt/C electrode (0.846 V). The limiting current was similar (~5.0 mA cm^−2^). The results confirmed that Co@CNTs-800 exhibited better ORR performance than Co@CNTs-600, Co@CNTs-700, and Co@CNTs-900. The ORR performance of Co@CNTs-800 is equal to previously reported non-precious ORR electrocatalysts (Appendix A). As shown in Appendix A, the half wave potentials were 0.767, 0.806, and 0.808 V vs. RHE for blank CNTs, Co-MOF-800, and Co-Zn-MOF-800, respectively, which were lower than that of Co@CNTs-800 (0.846 V vs. RHE). The results indicated that metal centers and the introduction of N played an important role in facilitating ORR. The electron transfer number and peroxide yield of the Co@CNTs-800 catalyst were calculated by the RRDE technique. In Figure 5b, the electron transfer number of Co@CNTs-800 was calculated as 3.94 and the H_2_O_2_ yield was only 2.63%, which indicated that the 4-electron transfer pathway played a leading role in ORR. The electrochemical stability testing of the Co@CNTs-800 catalyst was investigated by chronoamperometry methods at a potential of 0.7 V. As shown in Appendix A, the current of Co@CNTs-800 was more than 90% retention after a 12 h test. LSV curves (Appendix A) were also measured after the 12 h i-t test. Half wave potential remained almost unchanged, demonstrating the excellent stability of Co@CNTs- 800 as ORR catalyst in alkaline solution. We attributed the excellent stability to the carbon shell surrounding the crystalline Co nanoparticles acting as a protector [34].

The Tafel slope values were calculated to further demonstrate the reaction dynamics of the electrocatalysts as shown in Figure 5c, Tafel slope values of Co@CNTs-600, Co@CNTs-700, Co@CNTs-800, Co@CNTs-900, and Pt/C were calculated to be 118, 112, 111, 114, and 125 mV·dec^−1^ respectively, which indicated that Co@CNTs-800 showed the fastest kinetics for electrocatalytic ORR [49]. In addition, electrochemical impedance spectroscopy was recorded in 0.1 M KOH to determine their ORR behavior. Figure 5d shows the Nyquist plots of all the electrodes. The high frequency semicircles reflect the ion adsorption kinetics and charge transfer resistance (Rct). Co@CNTs-800 with a smaller semicircle showed faster ion adsorption kinetics and charge transfer than Co@CNTs-600, Co@CNTs-700, and Co@CNTs-900. The Warburg length for Co@CNTs-800 in the 45° slope lines region in the middle frequency indicated its faster ion diffusion and transport compared to Co@CNTs-600, Co@CNTs-700, and Co@CNTs-900 [50]. We concluded that the three-dimensional networks of Co@CNTs-800, formed out of N-doped nanotubes, were beneficial to charge transfer as well as ion diffusion and transport. Thus, Co@CNTs-800 exhibited superior electrochemical activity for ORR.

## 4. Conclusions

In summary, Co@CNTs-800 with a three-dimensional network structure was synthesized by pyrolysis of a mixture of Co-Zn MOF and DCD. A three-dimensional network was formed out of N-doped carbon nanotubes with Co nanoparticles embedded. Benefiting from the nano-tubular structure, the as-prepared Co@CNTs-800 catalyst exhibited fast charge transfer and ion diffusion. In addition, the carbon shell coated on the Co nanoparticles was conducive in enhancing the stability of ORR electrocatalysts. Besides, more defects were created by Zn volatilization and introduction of N by DCD. Thus, Co@CNTs-800 demonstrated superior electrochemical activity for ORR as well as long-term durability. This work provides a synthetic method to prepare efficient electrocatalysts with three-dimensional networks for ORR.

## Data Availability

Data is available on the request from the corresponding author.

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
