# Peer review of "Co-Zn-MOFs Derived N-Doped Carbon Nanotubes with Crystalline Co Nanoparticles Embedded as Effective Oxygen Electrocatalysts"

_nanomaterials, 2021, doi:10.3390/nano11020261_

Round 1
Reviewer 1 Report
In this work by Zhang et al., the authors have investigated how the transformation of Co-Zn-MOFs gives a material of interesting electrochemical performance. They hypothesize that these precursors generate carbon nanotubes. However, there are certain issues with the manuscript highlighted below, which should be clarified to make this work publishable. At present, the given arguments are not convincing and more work is required to support the hypotheses.
1) "fenton reaction" (Line 31) - should be capitalized
2) "Firstly, 0.75 mmol benzenedicarboxylic acid (H2BDC) was dissolved to 36 mL mixed solution of DMF, ethanol and
deionized water with volume ratio of 16:1:1." (Lines 55-57) - did you use phthalic acid, isophthalic acid, or terephthalic acid?
3) "After being stirred for 5 min the above mixture suffered 8 h ultrasonication (40 kHz)." (Lines 59-60) - besides the frequency and time report amplitude, which is most important
4) "Typically, the homogeneous black solution was formed by 3 mg sample of Co@CNTs-T, 7 mg carbon black and 1.25 mL ethanol with ultraphonic treatment for 3 h." (Lines 78-79) - what is ultraphonic treatment?
5) "Commercial Pt/C electrode was 82 prepared by the same method." (Lines 81-82) - please be more exact and report the conditions
6) Micrographs and plots in the main text and SI should have the uniform size to enable data analysis. Moreover, some of the plots are too small to read e.g. Fig. S1c.
7) It would be good to provide higher resolution images of the CNTs in Fig. 1. At present, the magnification is too low to judge how many CNTs are present in the material. Admittedly, a TEM micrograph is given but there is always a chance that a lone CNT was located and visualized. Please provide a TEM image showing more CNTs.
8) Where are the peaks from CNTs in Fig. 2? You said that "No other diffraction peaks were in the XRD patterns, which demonstrated that no other phases appeared. " (Lines 122-123) suggesting that only Co and ZnO are present in the material, but no CNTs. The experimental mistake was to measure XRD from 30 deg. angle because CNTs manifest their presence just below 30 degrees. Please remeasure the material in the extended range.
9) It is a common practice to report Raman spectra of nanocarbon materials - particularly the I(D)/I(G) ratio to enable evaluation of the purity of the material. The purity in turn makes a major impact on the properties of the material, which can be key to understanding the observed electrochemical performance. As a consequence, it would be vital to provide the results of such a characterization.
Reviewer 2 Report
Please see attached file for Reviewer's comments.

Reviewer 3 Report
In this work, the author synthesized crystalline cobalt-embedded N-doped carbon nanotubes (Co@CNTs-T) via facile carbonization of a Co/Zn metal-organic frameworks and dicyandiamide at different carbonization temperature. They found Co@CNTs-800 is best among all the materials for ORR catalyst. This work is a good piece of work and I can recommend this for publications after Major revision.
Please find my comments below.
- I am quite surprised the author did not highlight metal-free electrocatalyst for ORR. Though the author focused on Co-based carbon, it is good to dig the literature on noble, 3d metal, and metal-free electrocatalyst.
- I do not agree with the tern Nanotube. I believe these are Co dopped carbon not really nanotubes.
- I would appreciate it if the author provides a table of comparison of 10-20 ORR electrocatalyst as a comparison.
- I am quite surprised to see, Co@CNTs-X (X = 600-800) are non-porous (Figure 4).
- I would rather see the surface area properties of CoZn-BDC MOF. Please provide it with TGA analysis
- Please provide CHNS data for all Co@CNTs-X as a table.
- Please provide some discussion on why CoZn-BDC is used not only Co-BDC-MOF?
- I am wondering about the ORR of carbonized MOF without dicyandiamide? At least the author should check Co-MOF@800 for a comparative study.
- The author should provide the role of N (different N-species) on ORR. As a quick check, in this paper, ACS Appl. Mater. Interfaces 2020, 12, 40, 44689–44699, the author highlighted quaternary-N for ORR activity. The author should cite this paper and discuss more on the nature of N-species and their effect on ORR.
- The author should determine the Co/Zn content from ICP-MS analysis.
Round 2
Reviewer 1 Report
Thank you for the corrections. I recommend the publication of the article in the present form.
Reviewer 2 Report
The submitted manuscript deals with the preparation of Co-Zn-MOF derived N-doped carbon nanotubes with embedded Co nanoparticles as an efficient oxygen reduction reaction catalyst in alkaline media. Based on careful review of the revised version of the manuscript, it is clear that the authors have satisfactorily made necessary modifications to improve the overall quality of the manuscript, particularly with respect to the presentation of the characterization results, as well as providing necessary control data for the electrochemical evaluation. With this said, the manuscript can be accepted for publication in Nanomaterials without further modification.
Reviewer 3 Report
The authors modified their MS and implemented all the changes mentioned in my report. I am happy with the changes and now the manuscript looks great. I have only concern about the porosity but if the author already measured it a couple of times, I believe this material is non-porous.
I am happy to recommend this for publication.